# Temporal and Spatial Changes of Rural Settlements and Their Influencing Factors in Northeast China from 2000 to 2020

**Jieyong Wang [1,2,\*,†]** [iD], **Xiaoyang Wang [3,†], Guoming Du [3] and Haonan Zhang [1,2,4]**

1   Institute of Geographic Sciences and Natural Resources Research, Chinese Academy of Sciences, Beijing 100101, China
2   Key Laboratory of Regional Sustainable Development Modeling, Chinese Academy of Sciences, Beijing 100101, China
3   School of Public Administration and Law, Northeast Agricultural University, Harbin 150030, China
4   University of Chinese Academy of Sciences, Beijing 100049, China
\*   Correspondence: wjy@igsnrr.ac.cn
†   These authors contributed equally to this work.

**Abstract:** Rural settlements in Northeast China have undergone significant changes in the process of rapid urbanization, which has profoundly affected food production and the process of sustainable rural development. Based on multi-period remote sensing interpretation data of land use and economic statistics, this study quantitatively analyzes the temporal and spatial pattern change characteristics and influential factors of rural settlements in Northeast China from 2000 to 2020. The results show that: (i) Between 2000 and 2020, the area of rural settlements in Northeast China increased by 190,603.03 hectares, which accounts for 7.62% of the total rural settlements area, and 129 counties (cities) (70.88%) increased the area of rural settlements; (ii) The expanded state of rural settlements presents a low spatial distribution pattern in the northwest and high spatial distribution pattern in the southeast. The core density value of rural settlements in the north decreases, and the core density value in the southeast increases slightly. In addition, the landscape pattern of rural settlement expansion is irregular and there was increased disturbance from settlement expansion; (iii) A total of 81.6% of the land occupied by the expansion of rural settlements comes from cultivated land. The soil's organic matter content is 10.0 g/kg–20.0 g/kg, and the high-quality cultivated land occupied by the expansion is 218,274.17 hectares. However, it is interesting that the expansion of rural settlements coincides with the increasing number of hollow villages. From 2000 to 2020, the utilization degree of rural settlements in Northeast China decreased by 56.97%; (v) The main factors affecting the changes of rural settlements in Northeast China are water resource conditions, terrain conditions, traffic location, and the level of county economic development. In areas with superior agricultural production conditions, the influence of various factors on the change of rural settlements is more obvious.

**Keywords:** rural settlements; spatial and temporal variation; influencing factors; northeast region

## 1. Introduction

China has experienced a rapid and continuous process of urbanization since its reform and opening-up, which has led to the rapid rural-to-urban population migration [1,2]. Under the background of rapid and large-scale population urbanization, the relationship of man–land and the function of institutional structure in rural China have greatly changed [3–7]. Due to the lack of forward-looking planning and perfect land management systems in rural areas, "hollow villages" can be found everywhere and are especially serious in remote areas, and the phenomena of multi-family housing and newly built houses are common [8]. The rapid reduction of the rural population and the increase in settlements have resulted in a serious waste of land resources. Rural settlements are the core of rural production and life. The scale, structure, and form of rural settlements have always been

issues of concern to the international community, which results in significant changes in the rural geographical pattern. Explorations of the spatial and temporal patterns of rural settlements, the optimal layout of settlements, and the path model of sustainable development of villages have received extensive attention from scholars at home and abroad, and has become a research hotspot.

Rural settlements in China have been in a state of spontaneous evolution, and the number of hollow villages is increasing [5]. The future of rural settlements is unpredictable, but it can be explored. A significant amount research has been done on rural settlements abroad [9,10]. In recent years, several studies have been conducted on rural settlements, focusing on evolutionary characteristics of the temporal and spatial patterns of rural settlements, hollow villages, and the phenomenon of "reverse urbanization". The analysis of the spatial patterns of rural settlements mostly focuses on scale and structure changes, landscape morphology changes, influencing factors or driving mechanisms [11,12], etc. The spatial and temporal pattern evolution of rural settlements can be regarded as a reflection of the relationship between rural people and land [3,4], which reflects the changes in rural productivity and production relations [13–16]. Some studies have shown that the spatial and temporal patterns of rural settlements in China are changing in a complex way, along with the rapid urban expansion [17]. In order to meet the needs of rural development, the area of rural settlements is constantly expanding [18,19], and most of the expansion of rural settlements is the occupation of farmland on the fringes of rural settlements [20]. Due to external and internal drivers of change, the rural landscape is dynamic, continuous, and constantly changing [21]. The function of the countryside has changed, and the basic function of agricultural production has declined [22]. The changes in the structure and function of rural settlements and the increase in the intensity of human disturbance have resulted in heterogeneous changes in rural landscape morphology [23]. Changes in rural settlements are affected by many factors. The natural environment, traffic conditions, and economic development all affect the changes in the rural settlement pattern [20]. The natural environment plays a decisive role in the layout of rural settlements, but with advancements in science and technology, the influence of natural factors on the layout of rural settlements has weakened while the influence of social and economic factors has increased [24]. Comparing the results obtained by studying the spatial and temporal dynamics of rural settlements, the research on the influencing factors of rural settlements is relatively weak [20,25]. Most research on rural settlements focuses on the Yellow and Huang-Huai-Hai plains of China, the mountainous and hilly areas in the south [14,26], the Yangtze River Delta region [27], and the Beijing-Tianjin-Hebei region [28]. However, the spatial and temporal patterns of rural settlement changes in Northeast China have not been reported on in detail. The Northeast region, where urbanization started relatively early, is the main grain-producing area and industrial base in China where the rural population is decreasing the most and the fastest, so the problem of rural settlements is more prominent. Exploring the changing patterns of rural settlements in this region is important for reconfiguring rural space, optimizing rural industrial structure, and improving regional grain production capacity.

The purpose of this paper is to reveal the dynamic changes in the scale and landscape form of rural settlements in Northeast China, as well as the factors affecting the changes in rural settlements, and to propose targeted optimization paths and models. The article is divided into four parts: the first part explores the law of changes in the scale of rural settlements; the second part investigates the changes in the spatial morphology of rural settlements; the third part reveals the impact of dynamic changes in rural settlements, mainly through analyzing the occupied cultivated land and hollow villages; finally, the fourth part explores the factors affecting the changes in rural settlements in Northeast China. This study can identify the evolutionary characteristics of rural settlements in Northeast China and the loss of cultivated land quality and quantity due to changes in rural settlements, which is of great significance for guaranteeing national food security and protecting black land.

## 2. Materials and Methods

### 2.1. Study Area

Northeast China (115°E–135°E, 38°N–56°N) includes Liaoning Province, Jilin Province, Heilongjiang Province, Chifeng League, Tongliao League, Xing'an League, and Hulunbuir in eastern Inner Mongolia Autonomous Region. The Northeast region (Figure 1) has the largest plain in China of fertile black soil, and is in the three largest black soil regions in the world, which is very suitable for agricultural production. Northeast China is the largest commercial grain base in China. In 2020, the grain output was 17,346.88 tons, accounting for 25.91% of the country's total grain output. About 1/3 of the grain was transferred to other provinces. In addition, the Northeast region is also an old-fashioned industrial base in China, with a high level of urbanization and a serious loss of rural population. Compared to 2010, the rural population in Northeast China lost 14.56 million, and the population aging rate was 19.30%, which was 7.05% higher than in 2010 and 13.08% higher than in 2000. The population aging is worsening.

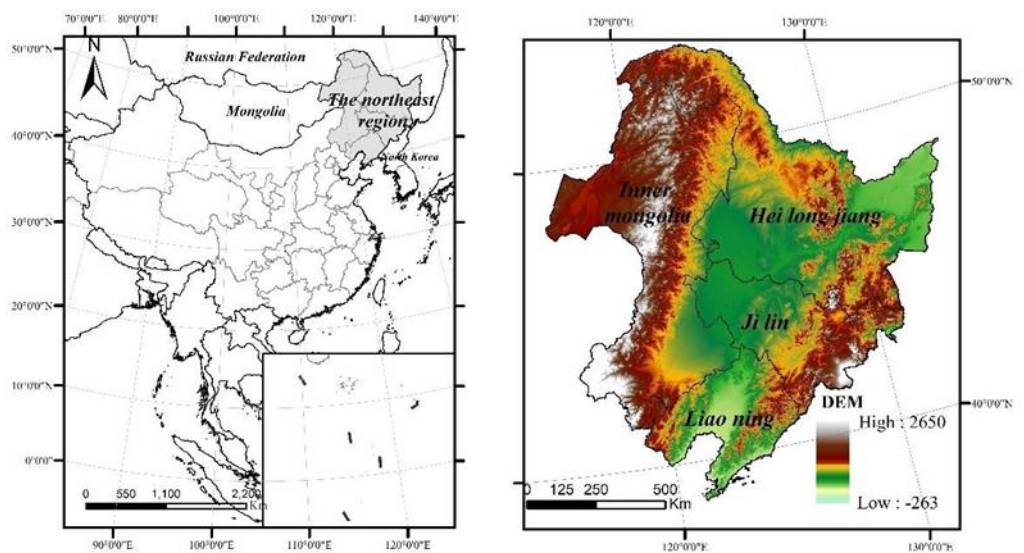

**Figure 1.** Map of the study area.

### 2.2. Data Source and Preprocessing

The data used in this study included: (i) Rural settlements data that were obtained from the remote sensing interpretation of land use by Resource and environmental science and data center, Chinese Academy of Sciences (http://www.resdc.cn/Datalist1.aspx (accessed on 15 February 2022)). This set of data divided land use into 6 categories and 25 subcategories (Table 1), and rural settlements are subcategories of urban and rural land, industrial and mining land, and residential land, which is independent of cities and towns [29]; (ii) soil organic matter data were obtained from the 1-km raster soil organic matter content map of China by the Institute of Soil Science, Chinese Academy of Sciences (http://soil.geodata.cn/ (accessed on 20 February 2022)); (iii) socioeconomic statistics, including the rural resident population and GDP per capita, were obtained from province and city statistical yearbooks (2000–2020) and statistical data released by the province and city statistics department. The land use data, soil organic matter data, and social statistics are released by the state. What we need to explain is that the sources of these data are uniformly obtained from the national level and the data are of the same standard to ensure the accuracy of the data.

<div align="center"><b>Table 1.</b> Land-Use and Land-Cover Change classification table.</div>

| First Class Type | | Secondary Type | | |
| --- | --- | --- | --- | --- |
| Numbering | Name | Numbering | Name | Meaning |
| 1 | Arable land | | | Land for planting crops, including cultivated land that has been in use, newly opened cultivated land, leisure land, land for crop rotation, and land for grass field rotation; agricultural fruit, mulberry, and agricultural and forestry land mainly used for planting crops; Beaches and tidal flats. |
| | | 11 | Paddy field | There are related facilities for water source guarantee and irrigation, generally irrigated arable land, arable land for growing aquatic crops such as rice and lotus root, and arable land where rice and dryland crops are planted in turn. |
| | | 12 | Dry land | There is no irrigation water source or facilities, and the arable land for growing aquatic crops depends on natural precipitation; the dry crop arable land that has water source and irrigation facilities and can be irrigated normally under normal conditions; the arable land mainly for vegetable cultivation; the idle land for crop rotation planting. |
| 2 | Woodland | | | Growing trees, shrubs, bamboos, and forestry land such as coastal mangroves. |
| | | 21 | High-density woodland | Refers to natural forests and plantations with canopy closure greater than 30%. |
| | | 22 | Bushland | Refers to low woodland and shrubland with canopy closure greater than 40% and height below 2 m. |
| | | 23 | Sparse woodland | Refers to forest land with a canopy density of 10–30%. |
| | | 24 | other woodland | Refers to undeveloped forest land, slashed land, nursery, and various gardens. |
| 3 | Grassland | | | All kinds of grasslands with a coverage of more than 5%, mainly of growing herbs, including nomadic shrub grasslands and sparse forest grasslands with a canopy closure of less than 10%. |
| | | 31 | High coverage grass | Natural grasslands, improved grasslands and mowing grasslands with coverage greater than 50%. This type of grassland generally has better water conditions, and the grass is densely grown. |
| | | 32 | Medium coverage grass | Natural grassland and improved grassland with 20–50% coverage. This type of grassland generally has better water conditions, and the grass is densely grown. |
| | | 33 | Low coverage grass | Natural grassland with 5–20% coverage. This type of grassland lacks water conditions, the grass is sparse, and the conditions for animal husbandry use are poor. |
| 4 | Waters | | | Natural land waters and land for water conservancy facilities |
| | | 41 | Canals | Naturally formed or artificially excavated rivers and land below the annual water level of the main trunk. |
| | | 42 | Lake | Naturally formed stagnant water area. The land below the perennial water level. |
| | | 43 | Reservoir pond | The land below the perennial water level in artificially constructed water storage areas. |

**Table 1.** *Cont.*

| First Class Type | | Secondary Type | | |
|---|---|---|---|---|
| Numbering | Name | Numbering | Name | Meaning |
| | | 44 | Permanent glacier snow | Land covered by glaciers and snow all year round. |
| | | 45 | Tidal flat | Tide flood zone between high tide level and low tide level of coastal spring tide. |
| | | 46 | Beach | The land between the water level of the river and lake during the flat-water period and the water level during the flood period. |
| 5 | Urban and rural construction land | | | Urban and rural residential areas and other lands for industry, mining, transportation, etc. |
| | | 51 | Urban land | Large, medium, and small cities and built-up areas above county and town |
| | | 52 | Rural settlement | Rural settlements independent of towns |
| | | 53 | Other construction land | Refers to factories and mines, large industrial areas and other land and traffic roads, airports, and special land. |
| 6 | Unused land | | | Unused land, including difficult-to-use land. |
| | | 61 | Sand | The surface is covered with sand, and the vegetation coverage is less than 5% of the land, including deserts but excluding deserts in water systems. |
| | | 62 | Gobi | The surface is dominated by crushed gravel, and the vegetation coverage is less than 5% of the land. |
| | | 63 | Saline-alkali land | The surface salt-alkali is concentrated, the vegetation is sparse, and only the soil with strong salt-alkali-tolerant plants can grow. |
| | | 64 | Wetlands | The terrain is flat and low-lying, with poor drainage, long-term humidity, seasonal or perennial water accumulation, and land with wet plants growing on the surface. |
| | | 65 | Bare earth | Land with surface soil coverage and vegetation coverage less than 5%. |
| | | 66 | Bare rock gravel | The surface is rock or gravel covering >5% of the land. |
| | | 67 | Other unused land | Other unused land, including alpine desert, tundra, etc. |

The main process of data processing is as follows: (i) Data pre-processing: this study adopted inverse distance weighted interpolation (IDW) to interpolate the number of the rural resident population for counties missing rural resident population data. As a commonly used interpolation method, IDW has a low average prediction error, so this paper selects the IDW interpolation method to predict the population of 13 missing counties and cities [30]. (ii) Data spatialization: With the support of ArcGIS, social and economic data as well as spatial data are correlated and superimposed to analyze the spatial characteristics of each influencing factor; this study uses related software to measure the morphological changes in rural homesteads. (iii) Statistical data analysis: SPSS 24 was applied to analyze the spatial information characteristics of rural residential land for statistical analysis of rural settlement changes.

*2.3. Research Methodology*

2.3.1. Spatio-Temporal Dynamic Characteristics of Rural Settlements

(1) Rural Settlement Expansion Index (RSI).

RSI is the ratio of the area of rural settlements at the end of the study period to the area of rural settlements at the beginning of the study period. RSI is generally applied to

reflect changes in the scale of rural settlements and to judge changes in the development of rural settlements. When RSI > 1, it indicates rural settlement area expanded; when RSI < 1, it indicates rural settlement area shrank [17]. The specific calculation formula is shown below.

$$RSI = RSA_F/RSA_B$$

where $RSA_F$ is the area of rural settlement land at the end of the study period and $RSA_B$ is the area of rural settlement land at the beginning of the study period.

(2) Kernel density estimation (KDE).

By creating a smooth circular surface for each feature point in the region, calculating the distance from the element point to the reference position based on a mathematical function, and summing all surfaces at the reference position, KDE builds a peak and kernel for these points to create a smooth continuous surface [31,32]. KDE is one of the statistical methods of nonparametric density estimation, which is modeled as follows.

$$f(x,y) = \frac{1}{nh^2} \sum_{i=1}^{n} k\left(\frac{d_i}{n}\right)$$

where f (x, y) is the density estimation located at (x, y) position, n is the observation numbers, h is the bandwidth or smoothing parameter, k is the kernel function, and $d_i$ is the distance from position (x, y) to observation position i.

(3) Landscape Shape Index (LSI).

Exploring the morphological changes of rural settlements by applying the method of large-scale integration (LSI). Calculate the landscape morphological index at the beginning and end of the research on rural settlements and analyze its dynamic change characteristics [14–16,33]. The mean shape index (MSI) and mean patch fractal dimension (MPFD) were selected to characterize the landscape shape changes. The higher the MSI value, the more complex the landscape shape; the higher the MPFD value, the greater the degree of human disturbance to the landscape shape.

a. Mean Shape Index (MSI).

$$MSI = \frac{\sum_{j=1}^{n} \frac{P_j}{\sqrt[2]{na_j}}}{n}$$

where $p_j$ is the perimeter of the rural settlement, $a_j$ is the area of the rural settlement, and n represents the number of rural settlements.

b. Average plaque typing dimension (MPFD).

$$MPFD = \sum_{i=1}^{m} \sum_{j=1}^{n} \left(\frac{2Ln\left(0.25p_{ij}\right)}{Lna_{ij}}\right) \frac{a_{ij}}{A}$$

where m is the number of patch types, n is the number of patches of a certain type, $p_{ij}$ is the perimeter of patch ij, $a_{ij}$ is the area of patch ij, and A is the area of the total landscape.

2.3.2. Implications of Rural Settlement Changes

(1) Impact of the changes in rural settlement on cultivated land (ROL)

This study uses the land transfer matrix to measure changes in rural settlements and arable land. ROL values are calculated based on the area of arable land occupied by rural settlement expansion and the content of soil organic matter. The land use conversion matrix (LUTM) is from the quantitative description of system states and state transfers in the system analysis [22,34]. The rows of the LUTM (Table 2) represent the land use types at time point $T_1$ and the columns represent the land use types at time point $T_2$. $P_{ij}$ represents the percentage of total land area converted from land type i to land type j during $T_1 - T_2$; $P_{ii}$ represents the percentage of the area where land use type i remains constant during

$T_1 - T_2$. $P_{i+}$ represents the percentage of the total area of land type i at time point $T_1$. $P_{+j}$ represents the percentage of the total area of land use type j at time point $T_2$.

**Table 2.** Land use transfer matrix [19].

| | | | $T_2$ | | | $P_{+i}$ | **Reduce Area** |
|---|---|---|---|---|---|---|---|
| | | $A_1$ | $A_2$ | ... | $A_n$ | | |
| | $A_1$ | $P_{11}$ | $P_{12}$ | ... | $P_{1n}$ | $P_{1+}$ | $P_{1+} - P_{11}$ |
| | $A_2$ | $P_{21}$ | $P_{22}$ | ... | $P_{2n}$ | $P_{2+}$ | $P_{2+} - P_{22}$ |
| $T_1$ | ... | ... | ... | ... | ... | ... | ... |
| | $A_n$ | $P_{n1}$ | $P_{n2}$ | ... | $P_{nn}$ | $P_{n+}$ | $P_{n+} - P_{nn}$ |
| $P_{+j}$ | | $P_{+1}$ | $P_{+2}$ | ... | $P_{+n}$ | 1 | |
| **Add area** | | $P_{+1} - P_{11}$ | $P_{+2} - P_{22}$ | ... | $P_{+n} - P_{nn}$ | | |

The ROL value reflects the characteristics of cultivated land occupied by rural settlements. When ROL < 0, the smaller the ROL value, the larger the area of cultivated land occupied by rural settlements and the more the soil organic matter; when ROL > 0, the larger the ROL value, the larger the area of rural settlements that reclaimed into cultivated land and the more soil organic matter there is. The specific formula is as follows:

$$ROL = \left[ P_{+n} - P_{nn} \times \sum_{i=1}^{m} D_i \right] \text{ or } ROL = \left[ (P_{n+} - P_{nn}) \times \sum_{i=1}^{m} D_i \right]$$

where $D_i$ is the soil organic matter class (m = 1,2...).

(2) Degree of population agglomeration of rural settlements (RHO).

RHO reflects the degree of hollowing out of rural settlements, which reflects the degree of rural population loss and the utilization efficiency of the rural homestead. When RHO > 1, it indicates that the degree of hollowing out of rural settlements is low, the utilization rate of the rural homestead is high, and the rural population loss is low; when RHO < 1, it indicates that the degree of hollowing out of rural settlements is high, the rural population loss is high, and the rural homestead waste rate is high [17]. The specific formula is as follows:

$$RHO = (POP_F / RSA_F) / (POP_B / RSA_B)$$

where $POP_F$ represents the rural population at the end of the study period and $POP_B$ represents the rural population at the beginning of the study period.

2.3.3. Analysis Model of Factors Influencing the Change of Rural Settlements

Rural settlements are influenced by a variety of factors such as natural ecology and economic technology [1]. The natural environment, traffic conditions, and economic development level affect the type and pattern of land use change in rural settlements [1,5], and the impact of economic development level is increasing [17]. Based on the distribution characteristics of rural settlements in Northeast China combined with field research, this study selects topographic conditions, location conditions, traffic conditions, water resource conditions, and county economic development levels as the main influencing factors. Topographical conditions are the basic elements that affect the distribution of rural settlements. Studies have shown that rural settlements are mostly distributed in flat areas such as plains and hills, and rural settlements in plain areas are more scattered, while rural settlements in mountainous areas are more concentrated [5]. The change in rural settlements is inseparable from the location conditions [7], the rural settlements closer to the city gave full play to the urban drive and promoted the development of rural industries [24]. This study constructs a multi-level buffer zone centered on cities and towns to measure the location conditions of rural settlements. Water resources are the basis for human production and living activities, and most of the traditional villages are built in water-rich areas such as by rivers and lakes [35]. For rural residents, water is vital to agricultural production and an important factor affecting crop yields. In recent years, the planting structure in Northeast China has changed significantly from dry land to paddy

field, and the demand for water resources has further increased. Therefore, the status of water resources is one of the important factors affecting the changes in rural settlements in Northeast China. Transportation condition is regarded as the basis for promoting economic development [17,26]. Road accessibility not only plays a major role in rural settlements but also promotes the secondary development of rural settlements [24]. Road accessibility drives changes in the structure of rural settlements. In this study, main traffic lines such as railways, national highways, and provincial highways are centered, and road buffer zones are constructed to evaluate the road accessibility of rural settlements. The level of economic development of the county is expressed by GDP per capita [26]. Rural settlements with higher GDP per capita tend to have better industrial development, relatively complete support facilities, larger populations, and larger rural settlements. On this basis, a linear regression equation affecting the change of regional rural settlements was constructed.

$$Ya = \beta0 + \beta1X1a + \beta2X2a + \beta3X3a + \beta4X4a + \beta5X5a + \varepsilon a$$

$Y_a$ is the index of rural settlement change from 2000 to 2020, a is the number of samples (a = 1.2. 3....n), $X_1$ is the topographic condition, $X_2$ is the locational condition, $X_3$ is the level of economic development, $X_4$ is the water condition, $X_5$ is the transportation condition, $\beta_0$. $\beta_1$.... $\beta_5$ are the undetermined parameters, and $\varepsilon_a$ is the random variables.

## 3. Results

### 3.1. Spatio-Temporal Dynamic Characteristics of Rural Settlements

3.1.1. Rural Settlement Expansion Characteristics

The area of rural settlements in Northeast China showed a significant increase from 2000 to 2020. The area increased by a total of 190,603.03 hectares, or 7.62%, over 20 years. There are 129 counties and cities (70.88%) with RSI values greater than 1, indicating an increase in the area of rural settlements. Seventy-one counties and cities (39.01) have RSI values greater than 1.14, indicating a large increase in the area of rural settlements. Eight counties and cities have RSI values greater than 2, indicating a doubling of the area of rural settlements. The RSI (Figure 2) values in the eastern and southwestern parts of the Liaohe Plain and the central part of the Sonnet Plain are larger, and the area of rural settlements has expanded significantly. The RSI values in the western part of the Northeast Plain are between 0.93 and 1.14, with relatively small changes, and the rural settlements are stable. The eastern part of the San Jiang Plain has the lowest RSI value, and the area of rural settlements has decreased considerably; the northern part of the Sonnet Plain has a lower RSI value, and the area of rural settlements has decreased slightly.

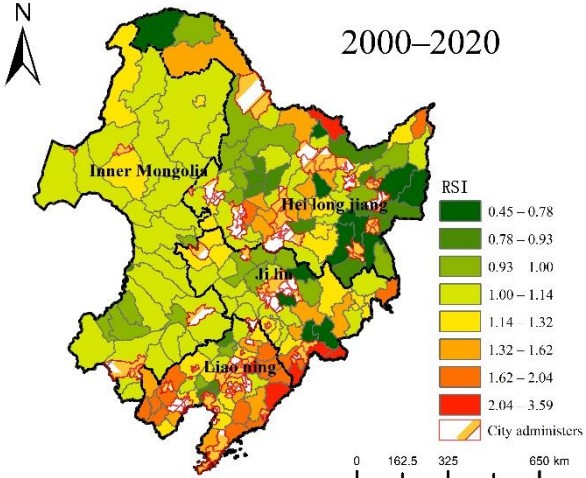

**Figure 2.** Spatial and temporal distribution pattern of rural settlement size in Northeast China.

In summary, the RSI values are larger in the central part of the plain and smaller in the peripheral areas. The RSI values in the peripheral areas of the cities are significantly higher than those in the other areas, and the area of rural settlements increased significantly due to urban radiation. The topography and proximity to the city may influence the evolution of rural settlements.

### 3.1.2. Spatial Distribution Characteristics of Rural Settlements and Their Changes

From 2000 to 2020, the density values (Figure 3) in some areas changed slightly, and the remaining areas were basically unchanged. The area of rural settlements in the Northeast Plain increased slightly, while the area of rural settlements in most areas decreased slightly, and the maximum decrease in nuclear density was only 0.02/km$^2$. The core density value of rural settlements in the central and western regions decreased, the maximum value of the decrease in the core density was 0.327/km$^2$, and the reduction rate of rural settlements increased. The core density value of rural settlements in the southeast region increased, and the area of rural settlements increased.

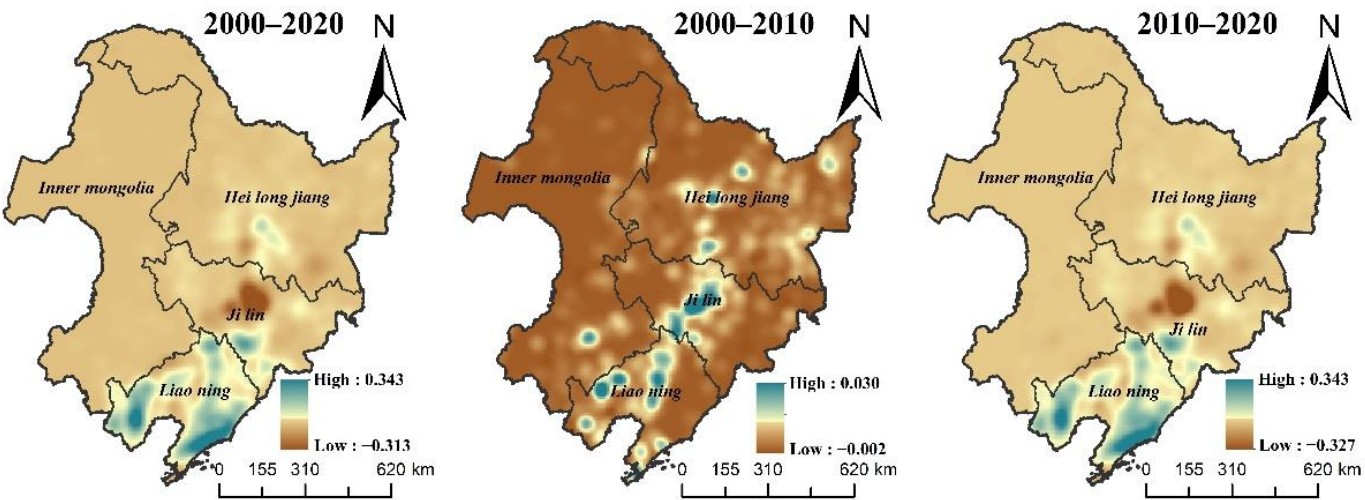

**Figure 3.** Variation of the nuclear density of rural settlements in Northeast China.

Overall, the core density value of rural settlements in the flat terrain increased slightly, and the increase in the core density value of rural settlements in water-rich areas was more obvious. Topography and water resources may influence changes in rural settlements.

### 3.1.3. Landscape Shape Change in Rural Settlements

The results show that from 2000 to 2020, the MSI and MPFD of rural settlements in Northeast China showed an upward trend, and the landscape morphology of rural settlements showed an irregular and complex development trend. Compared with other areas, the MSI and MPFD values (Figure 4) in the urban area increased significantly, and the development of rural settlements was more affected by human interference. The irregular expansion of rural settlements leads to the fragmentation of cultivated land patches around rural settlements, making the rural landscape pattern more complex. Therefore, the level of economic development may affect the changes in the landscape morphology of rural settlements.

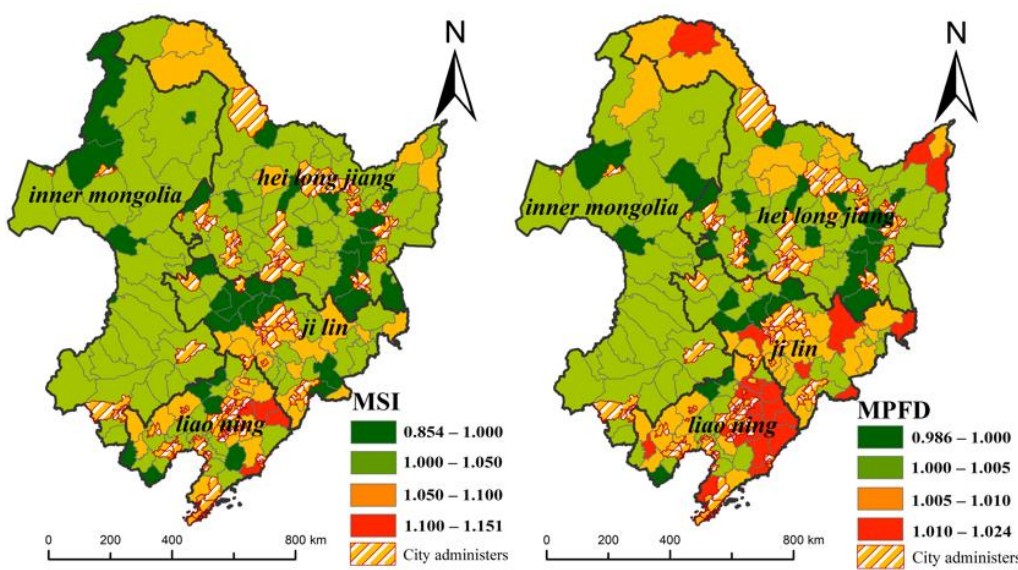

**Figure 4.** Landscape shape characteristics of rural settlements in Northeast China.

### 3.2. *Implications of Rural Settlement Changes*

3.2.1. Impact of the Changes in RURAL Settlement on Cultivated Land

From 2000 to 2020, changes in rural settlements were mainly concentrated in the central part of the Northeast Plain. The dynamic changes (Figure 5 a) in rural settlements are closely related to cultivated land. The land use types occupied by expansion (Table 3) are mainly cultivated land, accounting for 81.60%, of which dry land accounts for 92.30%, paddy fields account for 7.70%, and woodland, grassland, and water systems account for only a small proportion. The organic matter content of the expanded cultivated land was 96.88%, and the organic matter content was between 20 g/kg and 30 g/kg (Figure 5b), accounting for 2.31%. The organic matter content of the cultivated land occupied by the expansion of rural settlements was higher. Impact of the changes (Figure 6) in rural settlement on cultivated land (ROL) < 0 indicates that the cultivated land reclaimed by rural settlements cannot offset the impact of rural settlement expansion on cultivated land.

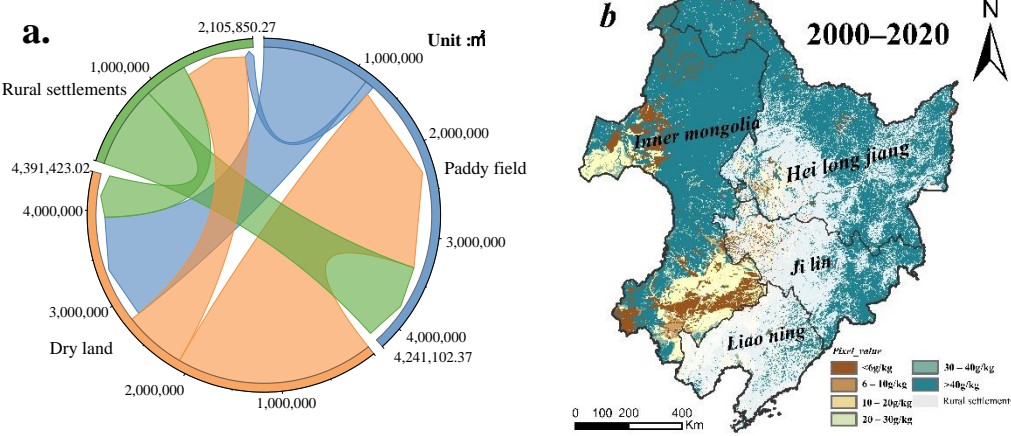

**Figure 5.** Characteristics of conversion of rural settlements and cultivated land: (**a**) dynamics of conversion of rural settlements and cultivated land area; (**b**) characteristics of cultivated land occupied by the expansion of rural settlements.

**Table 3.** Soil organic matter content of cultivated land occupied by the expansion of rural settlement.

| Type of Cultivated Land Occupied | Soil Organic Matter Content (g/kg) | Area (m²) | Proportion (%) |
|---|---|---|---|
| Paddy field | 10–20 | 15,777.47 | 7.17 |
| | 20–30 | 1150.96 | 0.52 |
| Dry land | <6 | 1520.40 | 0.69 |
| | 10–20 | 197,359.74 | 89.71 |
| | 20–30 | 3930.62 | 1.79 |
| | 30–40 | 55.38 | 0.03 |

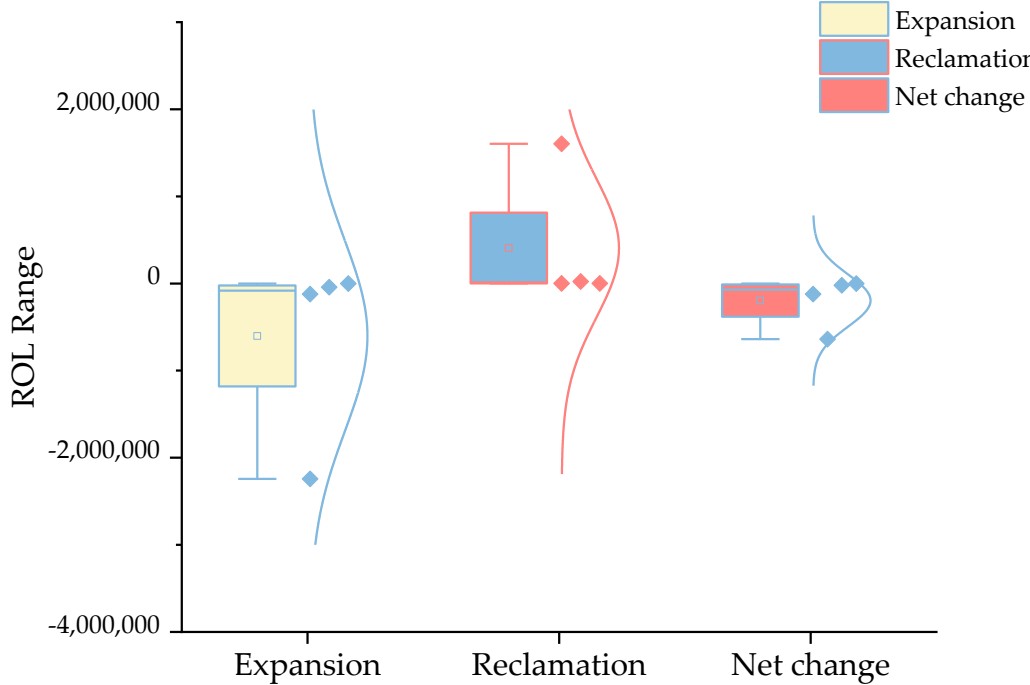

**Figure 6.** Statistics of the index of cultivated land occupied by rural settlements.

In summary, the conversion of rural settlements and cropland is in dynamic change. and the expansion of rural settlements not only has an impact on the quantity of cropland but also has an impact on the quality.

### 3.2.2. Population Agglomeration Characteristics of Rural Settlements

From 2000 to 2020, the rural population agglomeration in Northeast China has reduced, the rural population was drained seriously, and the utilization efficiency of the homestead was low. There are 141 (77.47%) counties in the Northeast with RHO (Figure 7) lower than the 1103 (56.59%) counties (cities) whose RHO is lower than 0.75. The spatial distribution pattern of RHO is high in the center and low in the south and north. Low RHO values appeared in the northern Songnen Plain, the eastern Sanjiang Plain, and the southwestern and eastern Liaohe Plain. The RHO value around the city is relatively high, driven by the radiation of the city, so the location conditions may affect the degree of hollowing out of the villages.

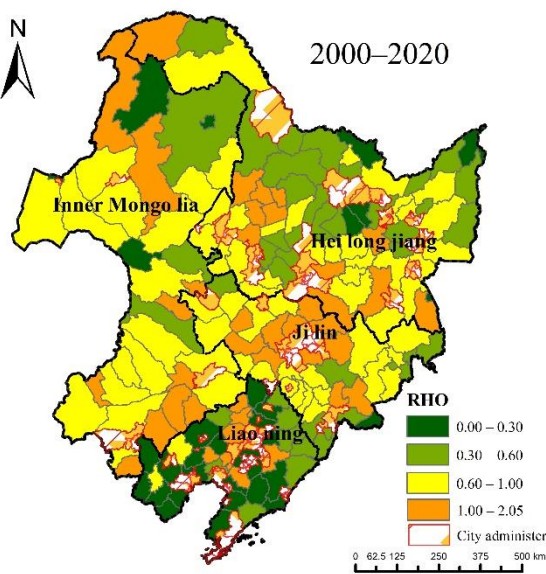

**Figure 7.** Spatial and temporal distribution of population clustering capacity in rural settlements in Northeast China.

### 3.3. Analysis of Factors Affecting Changes in Rural Settlements

Natural factors profoundly affect the structure and pattern of traditional rural settlements. Over time, rural economic development has become one of the important factors driving changes in rural settlements. [36]. This study uses a linear regression analysis model to analyze the factors that affect the changes in the spatiotemporal pattern of rural settlements in Northeast China from 2000 to 2020 and uses a random effects model (Table 4) for analysis. Random effects models were calibrated using the chi-square test (Sig). When the p-value is less than 0.05, the random effect can be rejected by transferring it to a fixed effects model.

**Table 4.** The relationship between landscape indicators and variables in rural settlements.

| Independent Factor | RSI | RCI | MSI | PDF | KDE |
|---|---|---|---|---|---|
| Location conditions | 0.131 | −0.048 | −0.01 | −0.019 | −0.023 |
| Water Resources Conditions | −0.215 ** | 0.209 ** | −0241 ** | −0273 ** | −0.018 |
| Terrain conditions | 0.015 | 0.244 ** | −0.102 | −0.092 | −0.03 |
| County Economic Development Level | −0.01 | −0.045 | −0166 * | −0.098 | −0.017 |
| Traffic conditions | 0.180 * | −0.058 | −0.016 | 0.017 | −0.055 |

** Significant correlation at the 0.01 level * at the 0.05 level (two-tailed) with significant correlation. (Two-tailed).

Water resources are closely related to agricultural production, life, and ecology, and are the main factor affecting rural settlement changes in Northeast China (Figure 8), with a 1% significant correlation with RSI, RCI, MSI, and MPFD. The northeastern region is China's commodity grain base, with arable land accounting for about 16% of the country's arable land area, and about 2/3 of the grain used for sale. It is a veritable "ballast stone" for Chinese grain. Water resources are the basic conditions for agricultural production. With the advancement of technology and facilities, a large amount of dry land in Northeast China has been transformed into paddy fields. In addition, from 2000 to 2020, the area of rural settlements in Northeast China reclaimed into paddy fields is twice the area reclaimed into dry land, which increases the demand for water resources. Areas rich in water resources can fully guarantee the domestic and rural industrial water needs of residents and lay the foundation for the development of rural industries.

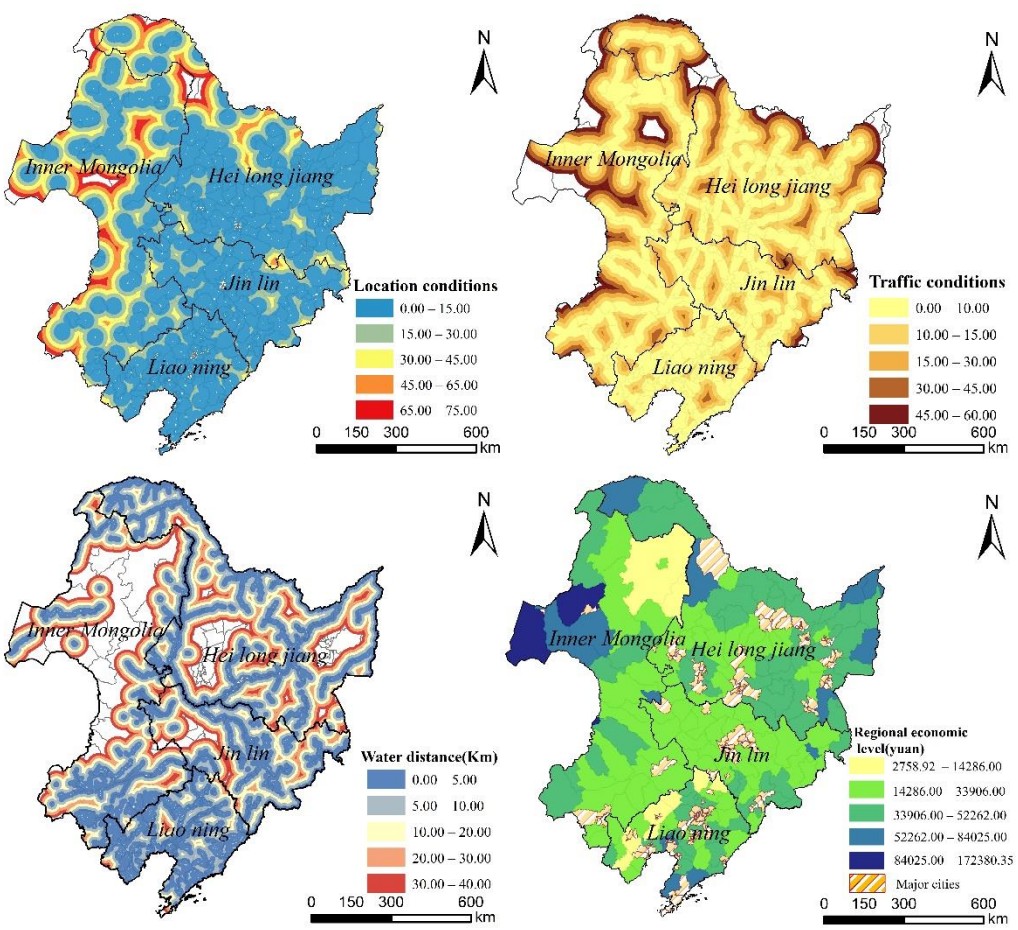

**Figure 8.** Spatial layout of each influencing factor.

Compared with the mountainous and hilly areas, the rural settlements in the plain area have a stronger ability to gather populations, and the traditional agricultural areas in Northeast China are in the plains. Relying on the development foundation of traditional agricultural resources, the expanding rural settlements are mainly concentrated in the flat plain areas. However, cultivated land in Northeast China, especially black land, is also concentrated in the Northeast Plain, which has the natural advantage of developing large-scale cultivated land management. With the promotion and development of large-scale agricultural machinery and drones, agriculture is developing in the direction of large-scale modernization. The expansion of rural settlements will erode high-quality arable land, cause arable land fragmentation, and hinder the large-scale operation of agricultural production.

Transportation is an important factor driving the evolution of the spatiotemporal pattern of rural settlements. Traditional rural settlements are generally laid out along rivers and roads. Today, transportation is still the main factor affecting the changes in rural settlements. A complete transportation network is a basic condition to ensure the quality of life and daily travel of residents, and it is also the minimum requirement for the existence and development of rural settlements. In the context of the rapid development of the global economy, the relatively closed space of traditional rural settlements has been broken, and communication between rural areas and the outside world has become increasingly close. The road network (Figure 8) is the most intuitive and basic way for rural settlements to communicate with the outside world. Rural settlements in areas with convenient transportation are more conducive to the introduction of enterprises and social capital, and to promoting the development of rural industries.

The level of county economic development has a close impact on the changes in the landscape form of rural settlements. The economic level (Figure 8) of the southern counties

in the Northeast region is relatively high, and the area of rural settlement expansion is relatively large. Counties with a higher level of economic development have more funds to carry out the transformation of rural settlements and to improve and optimize the landscape form and internal structure of rural settlements. Counties with a higher level of economic development have a higher level of rural industrial development, which can provide residents with more employment opportunities and public service facilities. Due to the relative lack of rural land management systems and policies, phenomena such as "multiple households" and hollow villages have appeared, and the expansion of rural settlements is in a disorderly state.

Location conditions (Figure 8) are an important factor affecting the changes in rural settlements. Driven by the radiation development of the city, the rural settlements around the cities have complete public service facilities and an excellent living environment, and the rural settlements have a large population and a large scale. The outward expansion of cities encroached on rural settlement land, gathered scattered rural settlements, and, to a certain extent, promoted the large-scale management of cultivated land and the concentration of the population. However, the transfer of the urban processing industry to the rural areas on the urban fringe has encroached on some rural land. By relying on the investment of enterprises, rural settlements have improved and optimized, and the scale of rural settlements has been increased.

## 4. Discussion

### 4.1. The Change of Rural Settlements in Northeast China Is a Complex Systematic Process

Northeast China is in the process of rapid urbanization, and urban and rural elements flow frequently. The constituent elements of rural settlements are in the process of constant change and development [25,37]. The migration of a large number of rural people to cities has resulted in idle or abandoned rural homesteads and a serious waste of land resources [38,39]. Rural population loss in Northeast China is serious, but the area of rural settlements is increasing. This abnormal phenomenon also occurs in other regions. [7,40]. The reasons for this abnormal phenomenon may be: (i) With the steady growth of economic income, residents pursue a higher quality of life, and many residents choose to expand their houses; (ii) Traditional rural settlements are limited by the radius of cultivation and alternative layouts with fewer options. With the development and progress of science and technology, the degree of restriction of the farming radius on the layout of rural settlements has decreased, and rural settlements are now more willing to expand into areas with abundant water resources, developed transportation, and better economies.

### 4.2. Changes in Northeast Rural Settlement Affect National Grain Production

Northeast China is China's commodity grain production base, which has important economic value and strategic significance. During the COVID-19 outbreak, some developing countries that need to rely on external resources to ensure national food security have faced enormous pressure [41]. The continuous expansion of urban and rural construction land in China has put a lot of pressure on maintaining the red line of arable land [36]. The proportion of cultivated land occupied in the expansion of rural settlements in China is 60% [42]. However, the problem of the loss of high-quality arable land due to the expansion of rural settlements has not been clarified. The organic matter content of cultivated land occupied by the expansion of rural settlements in Northeast China from 2000 to 2020 is high, and the disorderly expansion of rural settlements will lead to the fragmentation of cultivated land patches. The main reasons may be: (i) Both high-quality cultivated land and rural settlements in Northeast China are distributed in plain areas, and surrounding rural settlements are mostly high-quality cultivated land with high soil organic matter content; (ii) The landscape of rural settlements in Northeast China is greatly disturbed by human activities, and the expansion of rural settlements is in a disordered state, which results in more fragmented rural patches and hinders the large-scale operation of cultivated land. According to the preliminary estimates of the results of this study, the expansion of

rural settlements in Northeast China caused about 39 million kilograms of food production losses, and the changes in rural settlements affected national food production.

*4.3. Limitations and Future Application Prospects*

Rural constituent elements are complex and diverse, and changes in a single element have no significant impact on the expansion of rural settlements. Natural factors such as water resources and topography are the dominant factors affecting the changes in rural settlements in Northeast China, and economic factors also play an influential role [24]. China's urbanization rate will continue to grow for a long time, and the migration of the rural population to cities will remain the main trend [17,43]. The rural settlement pattern will inevitably change. The development trend of rural settlements in Northeast China is: (i) For the convenience of agricultural production and changes in planting structure, rural settlements expand to areas with rich water sources; (ii) By relying on the developmental foundation of traditional agricultural areas and the advantages of resources and environments, rural settlements in plain areas will expand, and the area of rural settlements in mountainous and hilly regions will decrease [17]; (iii) In pursuit of a convenient production and living environment, the area of rural settlements in areas with developed transportation will increase; (v) According to the development experience of developed countries, areas with a high level of economic development of rural settlements will increase. In the future, the rural population in Northeast China will still be in the process of transferring to cities. The large-scale and mechanized management of cultivated land is the development trend of agricultural production in Northeast China in the future.

This study has certain limitations that should be addressed in future studies. The survey found that rural settlements in Northeast China are expanding, but the population is rapidly decreasing. In-depth research found that the evolution of rural settlements in Northeast China is closely related to natural factors, especially agricultural production conditions. In addition, the factors affecting the change of rural settlements in Northeast China lack comprehensive consideration. In the future, the indicator data should be further refined to explore the driving mechanism of changes in rural settlements in Northeast China. Finally, this study found that changes in rural settlements in Northeast China will affect the quality and quantity of cultivated land. In future research, the impact of changes of rural residents in Northeast China on different agricultural types should be refined and the distribution pattern of rural settlements for agricultural modernization should be explored.

**5. Conclusions**

This study uses a fixed effects correlation model to analyze the temporal and spatial evolution characteristics of rural settlements in Northeast China and obtains the main factors affecting the changes in rural settlements in Northeast China. The research results show that: (i) From 2000 to 2020, the area of rural settlements in Northeast China increased by 190,603.03 hectares, and the expansion of rural settlements showed a low spatial distribution pattern in the northwest and high distribution pattern in the southeast. The kernel density value in the northern area of rural settlements decreased, and the kernel density value in the southeast area increased slightly. In addition, the development of rural settlements is greatly disturbed by human activities, and its landscape shape presents an irregular development trend; (ii) From 2000 to 2020, the hollowing out of rural areas in Northeast China increased by 56.97%, and the number of abandoned homesteads increased; (iii) Rural settlements among the land cover types were occupied by expansion, and cultivated land accounted for the largest proportion, approximately 81.60%, of which paddy fields accounted for 7.70% and dry land accounted for 92.30%. The organic matter content of most arable land is between 10 g/kg and 20 g/kg; (v) Water resource conditions, terrain conditions, traffic location, and the county economic development level are the main factors affecting the changes in rural settlements in Northeast China, and new rural

settlements in Northeast China are also mainly agglomerated in areas with abundant water sources and flat terrain.

According to the evolution characteristics and related influencing factors of rural settlements in Northeast China obtained in this study, the optimization of rural settlements in Northeast China should be carried out from four aspects. First, a strict cultivated land protection system should be established. [7] The construction of rural settlements should strictly control the occupation of cultivated land, especially high-quality cultivated land. Since China began to implement the strictest cultivated land protection system, a series of policies such as cultivated land balance and a permanent basic cultivated land protection system have been introduced, which have guaranteed the quantity and quality of cultivated land to a certain extent and prevented rural settlements from occupying high-quality cultivated land. The second is to optimize the village system, formulate rural development plans, compile practical village plans according to the existing city and county planning guidelines, determine the type of village development, and optimize the layout of village land to fully consider the impact of topographical conditions, water resources, location conditions, economic development level, and other factors on rural settlements. The third is to renovate hollow villages to improve land use efficiency. To explore the paid withdrawal mechanism of rural idle homesteads, the government should fully grasp the matching of regional human resources and land resources, rectify the rural idle homesteads, coordinate the contradiction between construction land and agricultural land, and improve the efficiency of land use. Finally, the fourth is to promote land transfer and scale management and build a suitable agricultural production system [44]. Northeast China has the inherent advantage of developing large-scale operations, and large-scale state-owned farms have also accumulated successful experience in the large-scale operation of cultivated land.

**Author Contributions:** Conceptualization, X.W. and J.W.; methodology, J.W. validation, J.W., G.D. and X.W.; formal analysis, X.W.; data curation, X.W. and H.Z.; writing—original draft preparation, X.W.; writing—review and editing, X.W., J.W.; visualization, X.W.; supervision, J.W. All authors have read and agreed to the published version of the manuscript.

**Funding:** This research was supported by the Strategic Priority Research Program of the Chinese Academy of Sciences (Grant No. XDA28130400), the National Natural Science Foundation of China (Grant No. 42171266).

**Institutional Review Board Statement:** Not applicable.

**Informed Consent Statement:** Not applicable.

**Data Availability Statement:** Data available in a publicly accessible repository.

**Acknowledgments:** We would like to thank the Center for Resource and Environmental Science and Data, Chinese Academy of Sciences (http://www.resdc.cn/Datalist1.aspx (accessed on 15 February 2022)) for the interpretation of land use remote sensing data and the Nanjing Institute of Soil Sciences, Chinese Academy of Sciences (http://soil.geodata.cn (accessed on 20 February 2022)) of China's 1 km raster soil organic matter data. In addition, we would like to thank anonymous reviewers for their valuable comments and suggestions to improve this paper.

**Conflicts of Interest:** The authors declare no conflict of interest.

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
