# Peer review of "Temporal and Spatial Changes of Rural Settlements and Their Influencing Factors in Northeast China from 2000 to 2020"

_land, doi:10.3390/land11101640_

Round 1

Reviewer 1 Report

The study investigated the changes in rural settlement and the reasons affecting this change. The subject of the study is suitable for the scope of the journal. The general structure of the article is well formed. But this requires some correction.

First of all, I could not see references in all the formulas used. MSI, MPFD, RSI, ROL formulas should be briefly referenced to all formulas.

Why was the IDW method used in the study? Could a different method be used? The method part of the study should be explained in more detail. For example, for which data we used IDW, how many points did we interpolsayoon.

Apart from this, perhaps presenting the data sources used as tables will increase the readability of the article.

6 categories and 25 sub-categories in line 120 can be briefly explained and given as a table.

I think your suggestions should be added to the conclusion section.

In addition, more references can be made to studies in this field, namely to the literature.

To summarize, if you write the methods you use more descriptively, you will increase the readability of the article.

Author Response

Dear Editors and Reviewers:

Thank you very much for allowing us to revise our manuscript entitled "Temporal and Spatial Changes of Rural Settlement in Northeast China and Its Influencing Factors, 2000-2020" (ID: land-1898761). We thank the editors and reviewers for their valuable time and insightful comments. These comments are very helpful for us to improve the quality of the manuscript. We carefully studied the comments and revised the manuscript one by one based on constructive comments. The main corrections to the paper and the responses to the reviewers' comments are detailed as follows:

Reviewers' comments to the authors:

Reviewer #1:

The study investigates changes in rural settlements and the factors that influence such changes. The research topic fits within the scope of the journal. The general structure of the article is good. But this needs some corrections.

Response: Thank you very much for your time and constructive comments. Based on your comments, we have carefully revised all parts of the paper, especially in the Introduction, Methods, and Discussion sections. The main modifications we made to the paper are listed below.

Comment 1:

First, I can't see the references in all the formulas used. MSI, MPFD, RSI, and ROL formulas should briefly refer to all formulas.

Response: Thanks for your constructive comments. We have added references to the chosen method, the RSI formula is referenced in line 148, the formulas for MSI and MPFD are referenced in lines 172-172, and the ROL formula is an extension of the LUTM method (lines 196-197).

Comment 2:

Why use the IDW method in research? Can a different approach be used? The methods part of the study should be explained in more detail. For example, for which data do we use IDW, and how many points do we interpolate?

Response: Thanks for your constructive comments. We make a statement on lines 135-137 of the article, clarifying why this paper chose the IDW method and how many points were inserted using IDW.

Comment 3:

In addition to that, perhaps presenting the data source used as a table would increase the readability of the article. The 6 categories and 25 subcategories in row 120 can be explained simply and given in the tabular form.

Response: Thanks for your constructive comments. Based on your valuable comments, we have added Table 1 in lines 132-133 of the text, detailing the classification.

Comment 4:

I think your suggestion should be added to the conclusion section.

Response: Thanks for your constructive comments. Based on your valuable comments, adjust the structure of the article and add the suggestions from the original conclusion to the conclusion section. Specific correlation measures are put forward for the optimization path of rural settlements in Northeast China.

Comment 5:

In addition, you can refer to the research in this field, that is, the literature.

Response: Thank you for your valuable comments. According to your opinion, we refer to authoritative authors in this field, such as Long H L, Liu Y S, and other famous scholars.

All in all, if you write the methods you use to be more descriptive, you will increase the readability of your article.

Response: Thank you for your valuable comments. According to your constructive comments, we have refined the use of the article, which is embodied in the selection and application of the method, making the article more readable.

Reviewer 2 Report

I thank the authors for giving me the opportunity to read their interesting work.

I highlight some possible additions:

1) the literature mainly refers to Chinese authors, it could be improved by supplementing it with more citations of international works

2) comparisons with other similar and comparable studies for other parts of the country or other areas outside China are lacking

3) the choice of variables used to justify the model should be better justified. Above all, variables such as cost of land, availability of capital, availability of labour are excluded; this choice should be justified

4) The analysis does not distinguish between types of agricultural production, or at least it is not clear whether they are homogeneous or not

5) The limitations of the research, the benefits for policy makers and possible future developments should be included in the conclusions.

Author Response

Dear Editors and Reviewers:

Thank you very much for giving us the opportunity to revise our manuscript entitled "Temporal and Spatial Changes of Rural Settlement in Northeast China and Its Influencing Factors, 2000-2020" (ID: land-1898761). We thank the editors and reviewers for their valuable time and insightful comments. These comments are very helpful for us to improve the quality of the manuscript. We carefully studied the comments and revised the manuscript one by one based on constructive comments. The main corrections to the paper and the responses to the reviewers' comments are detailed as follows:

This paper is interesting. The author proposes an analysis of the temporal and spatial evolution characteristics of rural settlements in Northeast China based on a fixed-effects correlation model and highlights the main factors affecting the changes in rural settlements in Northeast China.

This paper is related to the submission of a special issue entitled "Land Consolidation and Rural Revitalization".

The whole paper is well structured, and the author achieves the set goals.

Some key points have been noted and must be addressed for the paper on land to be published.

Response: Thank you for your valuable time and insightful comments. We have carefully addressed all the comments received point-by-point. We believe that all sections of our revised version have been substantially improved, and we hope that our extensive revisions meet with your approval.

  • Specify 6 major categories and 25 subcategories introduced in line 120, and the 25 subcategories in line 120 can be simply explained and given in the tabular form

Response: Thanks for your constructive comments. According to your suggestion, 6 categories and 25 sub-categories are reflected in table1, which gives a simple explanation for each type and gives the specific definition of each type of land use.

  • Improve the quality of the legends, especially in Figures 2, 5, and 7.

Response: Thank you for your valuable comments. According to your suggestions, the quality of Figure 2, Figure 5, and Figure 7 has been improved, making the pictures clearer and the boundaries more obvious.

  • Emphasize the limitations of this study. Integrate possible future developments of this research.

Response: Thank you very much for your constructive comments. Based on your suggestions, the limitations of this study and possible future directions are highlighted in the Discussion section of this article. See lines 446-476 for details.

Integrate references based on international literature reviews.

Response: Thank you very much for your suggestions. According to your suggestions, the international and domestic literature has been integrated, the development direction of rural settlement research has been expounded, and the articles of well-known experts in related fields have been referred to.

  • Improve the English quality of papers.

Response: Thank you for your detailed suggestion. According to your suggestion, some language and words of the article have been corrected to make the article easier to read.

Reviewer 3 Report

The paper is interesting. The authors propose a analysis based on a fixed-effects correlation model of the temporal and spatial evolution characteristics of rural settlements in Northeast China highlighting the main factors affecting the changes in rural settlements in Northeast China.

The paper is pertinent with the Special Issue entitled " Land Consolidation and Rural Revitalization " in which it was submitted.

The paper as a whole is well structured, and the authors reach the goal set.

Some critical points have been noted and will have to be resolved in order to make the paper on Land publishable.

·        Specify the 6 categories and 25 subcategories introduced at line 120.

·        Improve the quality of the legends of the figures in particular for Figure 2,5 and 7.

·        Highlight the limitations of this study.

·        Integrate possible future developments of this research.

·        Integrate references based on a review of international literature.

·        Improve the quality of the English language of the paper text.

Author Response

(The authors gave the same response as above.)
